# Prevalence and factors associated with pediatric HIV therapy failure in a tertiary hospital in Asmara, Eritrea: A 15-year retrospective cohort study

**Samuel Tekle Mengistu**[1,2]*, **Ghirmay Ghebrekidan Ghebremeskel**[1,2], **Oliver Okoth Achila**[3], **Miriam Berhane Abrehe**[4], **Samuel Fisseha Tewelde**[4], **Mahmud Mohammed Idris**[2,4], **Tsegereda Gebrehiwot Tikue**[2,4], **Araia Berhane Mesfin**[5]

**1** Nakfa Hospital, Ministry of Health Northern Red Sea Branch, Nakfa, Eritrea, **2** Orotta School of Medicine, Orotta College of Medicine and Health Sciences, Asmara, Eritrea, **3** Department of Allied Health Sciences, Orotta College of Medicine and Health Sciences, Asmara, Eritrea, **4** Department of Pediatrics and Child Health, Orotta National Referral and Teaching Hospital, Asmara, Eritrea, **5** National Communicable Disease Control Division, Ministry of Health, Asmara, Eritrea

* teklesam7@gmail.com

## Abstract

### Introduction

Treatment failure (TF) in HIV infected children is a major concern in resource-constrained settings in Sub-Saharan Africa (SSA). This study investigated the prevalence, incidence, and factors associated with first-line cART failure using the virologic (plasma viral load), immunologic and clinical criteria among HIV-infected children.

### Methods

A retrospective cohort study of children (<18 years of age on treatment for a period of > 6 months) enrolled in the pediatric HIV/AIDs treatment program at Orotta National Pediatric Referral Hospital from January 2005 to December 2020 was conducted. Data were summarized using percentages, medians (± interquartile range (IQR)), or mean ± standard deviation (SD). Where appropriate, Pearson Chi-Squire (χ2) tests or Fishers exacts test, Kaplan–Meier (KM) estimates, and unadjusted and adjusted Cox-proportional hazard regression models were employed.

### Results

Out of 724 children with at least 24 weeks' follow-up 279 experienced therapy failure (TF) making prevalence of 38.5% (95% CI 35–42.2) over a median follow-up of 72 months (IQR, 49–112 months), with a crude incidence of failure of 6.5 events per 100- person-years (95% CI 5.8–7.3). In the adjusted Cox proportional hazards model, independent factors of TF were suboptimal adherence (Adjusted Hazard Ratio (aHR) = 2.9, 95% CI 2.2–3.9, p < 0.001), cART backbone other than Zidovudine and Lamivudine (aHR = 1.6, 95% CI 1.1–2.2, p = 0.01), severe immunosuppression (aHR = 1.5, 95% CI 1–2.4, p = 0.04), wasting or

**Data Availability Statement:** All relevant data are within the paper and its Supporting Information file.

**Funding:** The author(s) received no specific funding for this work.

**Competing interests:** The authors have declared that no competing interests exist.

**Abbreviations:** 3TC, Lamivudine; ABC, Abacavir; AIDs, Acquired Immunodeficiency syndrome; AZT, Zidovudine; cART, Combined Anti-Retroviral Therapy; CD4, Cluster Designation; d4T, Stavudine; DNA, Deoxyribose Nucleic Acid; EFV, Efavirenz; HIV, Human Immunodeficiency Virus; FTC, Emtricitabine; HMIS, Health Management Information System; HR/AHR–Hazard Rate, Adjusted Hazard Rate; LTFU, Lost To Follow Up; NNRTI, Non-Nucleoside Reverse Transcriptase Inhibitors; NVP, Nevirapine; PYFU, person-year follow-up; SD, Standard deviation; TF, Therapy failure; VL, Viral load; WHO, World Health Organization.

weight for height z-score < -2 (aHR = 1.5, 95% CI 1.1–2.1, p = 0.02), late cART initiation calendar years (aHR = 1.15, 95% CI 1.1–1.3, p < 0.001), and older age at cART initiation (aHR = 1.01, 95% CI 1–1.02, p < 0.001).

## Conclusions

Seven in one hundred children on first-line cART are likely to develop TF every year. To address this problem, access to viral load tests, adherence support, integration nutritional care into the clinic, and research on factors associated with suboptimal adherence should be prioritized.

## 1. Introduction

The World Health Organization estimated that in 2021, about 1.7 million children aged 0–14 years were living with HIV infection worldwide, 90% of whom were from sub-Saharan Africa (SSA) [1]. In the focus countries, the number of new infections in pediatrics declined, from 240,000 [160,000–380,000] in 2010 to 130,000 [87,000–210,000] in 2021 [1]. According to this report, in these countries, 53% [36–64%] of children aged 0–14 years living with HIV were on combined antiretroviral therapy (cART) [1] resulting in a marked decline in HIV/AIDS-related hospitalization and deaths. However, cART coverage for children living with HIV (CLHIV) in SSA remains behind that of adults [2]. Highlighting these concerns, a 2019 Joint United Nations Programme on HIV/AIDS (UNAIDS) noted that CLHIV are largely overlooked in HIV treatment scale-up programs and are not promptly treated and diagnosed early enough to prevent HIV-related morbidity and mortality [3]. In Eritrea, a 2019 Spectrum modeling estimated that the magnitude of people living with HIV/AIDS (PLWHA) was 14,000 (0.36% of the total population) and 8,956 (73%) patients are currently cART.

CLHIV (<18 years) make up 4% of PLWHA.

Multiple factors have been invoked to explain the treatment gap in children. In particular, there is a consensus among investigators that existing diagnostic and treatment approaches for children are complex and difficult to implement in resource-limited settings. Other lingering difficulties include unavailability of an age-appropriate treatment regimen; frequent co-infections; the wide use of drugs with low-genetic barriers to resistance; variable pharmacokinetics; suboptimal adherence; limited real-time viral load (VL) monitoring; high frequency (>10%) of pretreatment HIV drug resistance mutations (PDRMs) [4]; drug-related adverse events and drug stock-outs due to breaking downs in supply chains [5–9]. Besides, complex psychosocial problems such as caregiver support, over-centralization, and/or inappropriate integration of HIV/AIDS services into the broader child health platform further aggravate the problem [3]. Lastly, data suggest that a significant number of children (>50%) are exposed to suboptimal regimens, potentially leading to sub therapeutic drug concentrations [10].

An upshot of this catalog of barriers is a high drug failure rate in children. In general, studies in Low- and Middle-income Countries (LMIC) have reported TF rates of 10–34% among children after 2–3 years of cART [11]. In SSA, estimates of virologic failure (VF), with or without resistance mutations in children, range between 13–56% [12–15]. Also, delays in detecting early TF and subsequent switching to second-line therapy may compromise overall treatment outcomes [7, 16]. This is particularly relevant for children where such delays are common and are associated with increased risk of clinical progression to AIDS and higher morbidity and mortality [8].

At present, identifying and managing drivers of TF in children is a continuing concern [17]. However, the problem is not well studied in most resources constrained countries including Eritrea. Therefore, this study explores the frequency of pediatric HIV TF and its associated factors in one of the largest treatment centers in Eritrea.

## 2. Materials and methods

### 2.1 Study design and setting

We conducted a retrospective cohort study at the pediatric HIV/AIDs follow-up clinic in National Pediatric Referral Hospital (NPRH). HIV/AIDs follow-up clinic in NPRH was commissioned in 2005, making it the first institution in Eritrea to offer cART to CLHIV. Before the decentralization of services to other zones (2010), NPRH (in the Maekel zone) was the only institution offering cART to CLHIV in the country. In total, 822 children under the age of 18 years received service/or have been enrolled at the clinic since its inception.

Ethical approval was obtained from the Ministry of Health (MOH) research Ethics and Protocol review committee with a letter of reference: 21/09/20. All data collected from the records of respondents will be kept confidential at all times. Consent to participate was not obtained from the patients or guardians because the study was based on anonymized patient records and this was waived by the ethical committee. All procedures of the study also followed the recommendation of the Declaration of Helsinki Convention.

### 2.2 Study cohort description

All individuals ≤ 18 years old living with HIV/AIDS who attended the NPRH HIV follow-up clinic from 2005–2020 were enrolled in the study. Those with a follow-up duration of < 6 months missing key follow-up data, unknown follow-up duration, and those with no therapy outcome endpoints, were excluded (Fig 1). Unknown duration of follow-up refers to patients who don't have last point of their show-up in the clinic after being enrolled. It is treated differently from those who are known to be lost to follow-up. As those who are lost to follow-up could be censored according to their therapy outcome till the last point they were known to be in follow up. All eligible children and adolescents received free treatment for HIV–according to the national ART Guidelines. The guidelines endorsed the use of two Nucleoside Reverse Transcriptase Inhibitors (NRTIs) and a Non-Nucleoside Reverse Transcriptase Inhibitors NRTI (NNRTI) as the standard first-line regimen and use of protease inhibitors as second-line regimens. All Children were prescribed fixed-dose combination tablets. Assessment of drug adherence was routinely conducted by monitoring missed doses.

### 2.3 Data collection

Relevant data were extracted from an existing database and patients' clinical cards. Accordingly, all clinical cards were reviewed for demographic information, clinical, laboratory, and anthropometric data. The process was undertaken by trained health professionals from 12th September 2020 to 2nd February 2021 and was closely monitored by the principal investigator and supervisor.

### 2.4 Operational definitions

A. A case of TF is defined as, a patient who fulfills any definition of TF and/or has been switched to a second line due to TF with adherence support [6].

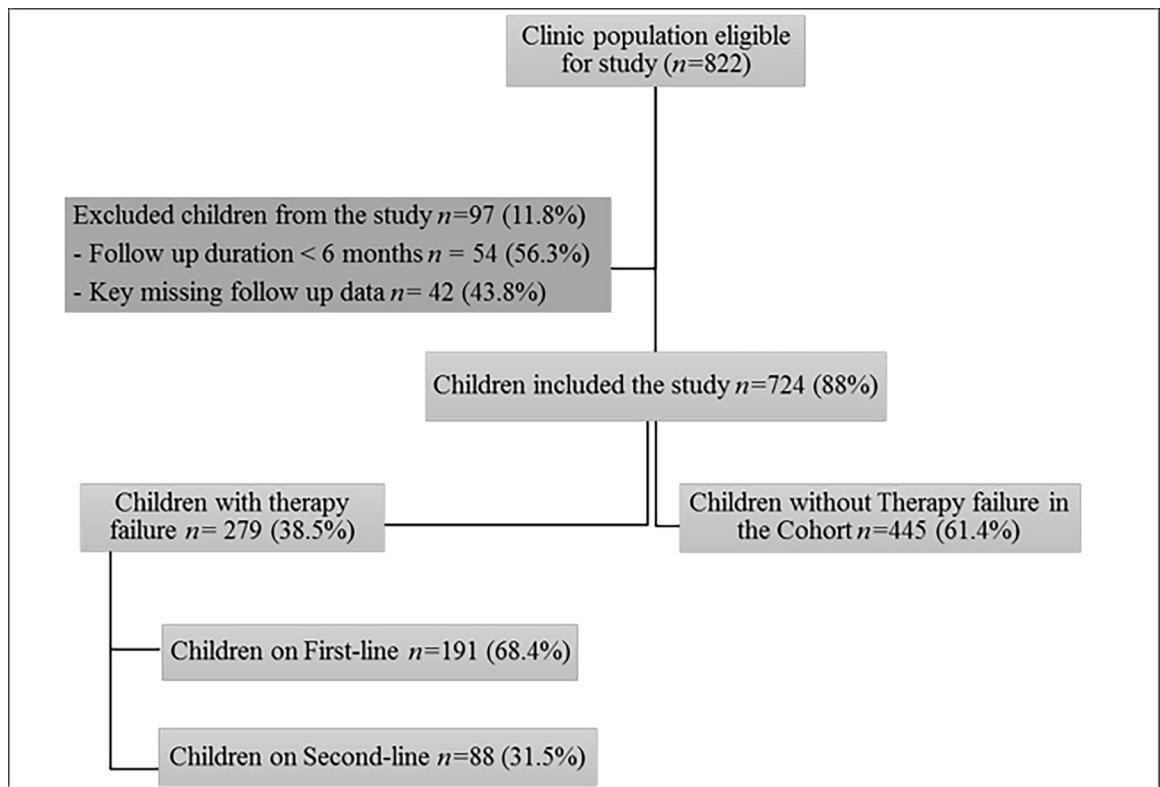

**Fig 1. Flow diagram of study recruitment and outcomes in children receiving cART in National Referral Pediatric Follow-up Clinic, 2005–2020.**

- Clinical TF: New or recurrent clinical event indicating advanced or severe immunodeficiency (WHO clinical stage 3 and 4 clinical conditions except for TB) after 6 months of treatment.

- Immunologic TF: In children <5 years, persistent $CD4^+$ levels <200 cells/mm$^3$ (count less than the threshold for two consecutive measures 6 months apart) and in children >5 years, persistent $CD4^+$ levels <100 cells/mm$^3$ despite treatment for 6 months.

- Virologic TF: plasma VL >1000 copies/ml based on two consecutive VL measurements of 3 months interval.

B. B. Adherence was assessed by pill count at each follow-up visit as good, fair, and poor if a child missed <5%, 5 to <10%, and >10% doses respectively of the expected monthly doses. Suboptimal adherence comprises of any patient with at least one record of fair or poor adherence. Adherence was categorized as maladherent or good adherent due the fact that each patient has separate column of adherence record for each follow up. Adherence data was available as a separate column for each follow up. Therefore, it was possible to summarize adherence profile from multiple time points.

C. C. Immunodeficiency is classified as mild, advanced, and severe according to the WHO 2016 thresholds [16].

D. D. WHO standard deviations (SD) for growth monitoring were calculated to define nutritional status of participants. Accordingly; Undernutrition was defined as "underweight"

(weight-for-age <−2SD); "stunting" (height-for-age <−2SD); "wasting" (weight-for-height <−2SD).

E.  E. Adolescents are individuals whose age is greater than 10 and less than 18 years. Children are those less than 10 years.

### 2.5 Endpoints definition

For TF analysis, the period of follow-up was from cART initiation up to the earliest detection of TF. Children without TF were censored at the date of death, lost to follow-up (defined as missing follow-up visits for more than 6 months), transferred to another clinic, or the date record of any last event in the clinic.

### 2.6 Statistical analyses

All analyses were conducted using SPSS version 26 (SPSS Inc., Chicago, Illinois, USA.) and Stata version 12.0 (Stata Corporation, College Station, TX). Where appropriate, demographic and HIV-related characteristics of patients were summarized using percentages, medians (± interquartile range (IQR)), or mean ± standard deviation (SD). Descriptive analyses were stratified by therapy outcome in all key variables at baseline using Pearson's Chi-square test or Fisher's exact test, and Mann-Whitney U test for continuous data. The incidence rate of TF was calculated by dividing the number of patients with TF by the total number of person-years of follow-up. Kaplan–Meier estimates and log-rank tests were performed to compare the cumulative incidence of TF between different categories of patient-specific characteristics. The confidence interval for incidence rates were calculated in Stata (StataCorp. 2011. Stata: Release 12. Statistical Software. College Station, TX: StataCorp LP) using the following method after defining the data in time to event form: main tab statistics > survival analysis > Summary statistics, tests and tables > Person-time, incidence rates and SMR.

Finally, Cox proportional hazards analysis was conducted to identify the factors associated with VF. The following variables were considered in a multivariate-adjusted Cox proportional hazards model of TF: initial cART treatment regimen, age at treatment initiation, adherence, gender, baseline disease stage, immunodeficiency status, frequency of cART changes, year od cART start and baseline anthropometric values for age z-score [18]. Adjusted Cox proportional hazards models were used to determine the odds of TF. Log-likelihood ratio was used to create these adjusted models, with a variable being included in the model if it resulted in an improvement in the model fit. Two-sided p-values < 0.05 were accepted as statistically significant.

## 3. Results

### 3.1 Demographic and clinical characteristics

A total of 822 CLHIV were screened for eligibility and 724(88%) fulfilled the inclusion criteria (Fig 1). Those eligible, with at least 24 weeks of follow-up, were followed for a total of 3913 person-years. In this period, 34/724 (4.7%) died, 105(14.5%) were lost to follow-up, and 403 (55.7%) were transferred out. The median age at enrollment to the clinic was 78 months (IQR, 38.5–114.5 months). Females comprise 47.2% of the population. Half of the children (50.1%) initiated cART before 2010, whereas 34.7% initiated between 2011 and 2015, and 15.2% in 2016 and onwards. More than half of the children (52%) had advanced HIV disease (WHO clinical stage 3 and 4). The median duration of follow-up was 79 months (IQR, 49–112 months).The excluded 97 children were majorly due missing key time to event data. Those

eligible were more likely to be from the Maekel zone (p-value = 0.03), in an advanced WHO clinical stage (p-value < 0.001), and on zidovudine (AZT) + lamivudine (3TC) backbone than their counterparts. Furthermore, the fewer of the study participants were wasted (weight for height, Z<-2) (p-value = 0.02), acutely malnourished (weight for age Z <-3) (p-value <0.001), and severely immunosuppressed (p-value = 0.037) when compared to eligible participants (Table 1).

### 3.2 Crude-incidence of cART failure

Crude incidence rates by key characteristics are provided in Table 2. A total of 279 TF events occurred. The prevalence of failure was 38.5% (95% CI 35–42.2). The crude incidence of TF was 6.5 events per 100-persons years of follow up (95% CI 5.8–7.3). The median time to TF was 4 years (IQR 2–7 years) among those who failed the first-line of therapy. Virologic failure (VF) occurred alone in 167/279 (23.1% [95% CI 20–26.3]) patients, immunologic failure (IF) alone was found in 19 (2.6% [95% CI 1.6–4.1]) cases, 82 (11.3% [95% CI 9.1–13.9]) had only clinical failure (CF) and 11 (1.5% [95% CI 0.8–2.7]) children had concomitant virologic, clinical and immunologic failures. Among all the children with TF, only 88 (31.5%) were switched to second-line treatments with the median time to cART switch being 19 months (IQR 11.3–49.7).

### 3.3 Prevalence and factors associated with TF

The children who failed on a first-line cART as compared to those who didn't were more likely to be in advanced WHO clinical stage (p = 0.005), have lower baseline $CD4^+$ lymphocyte count (p = 0.004), be stunted (P = 0.01), have had a longer duration of follow up (p = 0.003), have suboptimal adherence (p<0.001), have a record of cART change (p = 0.002) and have a higher frequency of regimen change (p = 0.001) (Table 3).

### 3.4 Relationship between number viral load tests performed and detection of TF

A strong relationship was observed between the number of VL tests performed and TF. A steep drop in the recruitment of infected children into the cART program and an increase in VL testing were also observed in the later years (Fig 2).

### 3.5 Kaplan-Meier analysis for TF incidence

The Kaplan-Meier estimates (Fig 3) of failure incidence comparing TF for the following factors; adherence, age at treatment initiation, cohort year, and cART backbone using median time to therapy failure. Sub-optimal adherence was associated with a reduced median time to TF (75.8 (95% CI, 65.7–85.96) months vs 117.2 (95% CI, 111.6–122.8) months) (Fig 3A). A significant difference in failure rates was also observed between adolescents and children: children, median = 120.7 (95% CI, 115–126.7) months; and adolescents, median = 78 (95% CI, 71.3–85) months (Fig 3B). Late initiation year were also associated with shorter median time to TF (2005–2009, mean = 124.85 (95% CI, 118.03–131.7) months; 2010–2014, median = 93.24 (95% CI, 88.3–98.1); 2015–2019, median 51.02 (95% CI, 47.1–54.95) (Fig 3C). The median time to TF for AZT+3TC was 114.5 (95%CI, 109.1–119.8) months vs 92.1 (95%CI, 79.7–104.4) months for alternative backbones (Fig 3D).

**Table 1. Baseline characteristics of HIV-infected children and adolescents in the in National Referral Pediatric Follow-up Clinic, Asmara, Eritrea (2005–2020).**

| Characteristics[†] | Included (n = 724) | Excluded (n = 97) | Total n (%) |
|---|---|---|---|
| **Gender** | | | |
| Male | 376 (52.7) | 63 (57.7) | 439 (53.4) |
| Female | 337 (47.3) | 46 (42.3) | 383 (46.6) |
| **Year of birth** | 2003 (1999–2005) | 2002 (1999–2006) | |
| **Address** | | | |
| Maekel | 523 (73.3) | 69 (63.8) | 592 (72) |
| Outside Maekel | 190 (26.7) | 40 (36.6) | 230 (28) |
| **cART initiation year** | 2011 (2008–2014) | 2010 (2007–2014) | |
| 2005–2009 | 274 (38.5) | 42 (39.2) | 316 (38.6) |
| 2010–2014 | 275 (38.6) | 44 (41.1) | 319 (62.2) |
| 2015–2019 | 162 (22.7) | 21 (19.6) | 183 (22.4) |
| **Age at cART initiation** | 105 (59–142) | 101 (68–135) | |
| < 67 months | 174 (24.4) | 31 (28.4) | 205 (25.1) |
| 67–102 months | 189 (26.5) | 19 (17.4) | 205 (25.1) |
| 103–136 months | 176 (24.7) | 30 (27.5) | 207 (25.3) |
| > 136 months | 172 (24.1) | 29 (26.6) | 201 (24.6) |
| Clinical stage[§] | | | |
| Early Stage | 291 (41.3) | 25 (23.1) | 316 (38.9) |
| Advanced Stage | 413 (58.7) | 83 (76.9) | 496 (61.1) |
| **TB Status** | | | |
| Not Symptomatic | 636 (97.5) | 88 (97.7) | 724 (97.6) |
| Symptomatic | 16 (2.5) | 2 (2.3) | 18 (2.4) |
| **cART backbone** | | | |
| AZT + 3TC | 539 (75.8) | 41 (37.9) | 580 (70.7) |
| ABC + 3TC | 89 (12.5) | 17 (15.7) | 106 (12.9) |
| D4T + 3TC | 49 (6.8) | 45 (41.6) | 94 (11.5) |
| TDF + FTC | 34 (4.7) | 5 (4.6) | 39 (4.8) |
| NNRTI/PI | | | |
| NVP | 360 (50.7) | 63 (58.8) | 423 (51.7) |
| EFV | 349 (48.3) | 44 (41.2) | 393 (48) |
| **Height for age, z-score** | | | |
| Z ≤ -2 | 470 (67.5) | 66 (74.1) | 536 (68.3) |
| Z > -2 | 226 (32.4) | 23 (25.9) | 249 (31.7) |
| **Weight for height, z-score** | | | |
| Z ≤ -2 | 40 (25.6) | 13 (56.5) | 53 (29.6) |
| Z > -2 | 116 (74.4) | 10 (44.5) | 126 (70.4) |
| **Weight for age, z-score** | | | |
| Z ≤ -2 | 309 (67.7) | 48 (92.3) | 357 (69.6) |
| Z > -2 | 148 (32.3) | 8 (7.7) | 156 (30.4) |
| **cART changes** | | | |
| Yes | 580 (79.4) | 21 (19.2) | 601 (73.1) |
| No | 124 (16.9) | 72 (66) | 196 (23.8) |
| Unknown | 9 (1.2) | 16 (16.8) | 25 (3) |
| **Immunosuppression** | | | |
| Mild | 87 (14.3) | 9 (11.3) | 96 (13.9) |
| Advanced | 201 (33.1) | 17 (21.5) | 224 (32.4) |

(*Continued*)

**Table 1.** (Continued)

| Characteristics[†] | Included (n = 724) | Excluded (n = 97) | Total n (%) |
|---|---|---|---|
| Severe | 319 (52.5) | 53 (67) | 372 (53.8) |

Abbreviations: ABC, abacavir; AZT, zidovudine; cART, combined antiretroviral therapy, CI, confidence interval; d4T, Stavudine; EFV, Efavirenz; IQR, interquartile range; LPV/r; NVP, nevirapine; TB, tuberculosis; 3TC, lamivudine; Z-score, NCHS standard deviation.

Superscripts

[†]Presented as n (%) for categorical data and median (interquartile range) for continuous data

[§]WHO clinical early and advanced refer to stage 1 & 2 and stages 3 &4, respectively.

**Description:** cART change: Change constitutes changes of cART regimen among the first line options due a reason other than therapy failure e.g., due to toxicity, drug-drug interaction, among other.

### 3.6 Multivariate analysis of independent factors of TF

In the adjusted Cox proportional hazards model independent factors of TF were suboptimal adherence (aHR = 2.9, 95%CI 2.2–3.9, p<0.001), cART backbone other than AZT and 3TC (aHR = 1.6, 95% CI 1.1–2.2, p = 0.01), severe immunosuppression (aHR = 1.5, 95%CI 1–2.4, p = 0.04), wasting or weight for height z ≤ -2 (aHR = 1.5, 95% CI 1.1–2.1, p = 0.02), late cART initiation calendar years (aHR = 1.15, 95%CI 1.1–1.3, p < 0.001), and older age at cART initiation (aHR = 1.01, 95%CI 1–1.02, p<0.001) (Table 4).

## 4. Discussion

This study documented the pediatric HIV TF rate in one of the largest referral facilities in Eritrea. In this cohort, the HIV TF rate was 38.5% (95% CI, 35–42.2) with a median (IQR) time to TF of 48 (IQR, 24–84) months. The results in this study are comparable to first-line NNRTI-based cART failure rate results elsewhere in the sub-continent– 34% in Kenya [9]; 29% (95% CI, 6–33) in Mozambique, and 34% (median of 26.4 months) in Uganda [19]. Perhaps, more importantly, a meta-analysis conducted in specific LMIC reported a pooled TF rate of 26–36% [11]. Admittedly, TF rates vary widely with relatively high and low failure rates reported in some jurisdictions in SSA –60% in the Central Africa Republic [12] vs. 14.1% in Ethiopia [14, 15]. The observed variation is largely influenced by study type, treatment duration, VL thresholds, and CD 4[+] T cell count thresholds employed, among others [15, 16, 20]. In general, studies deploying low CD4[+] T cell count (<50 cells/mm$^3$) or VL thresholds (<1000 copies/ml) tend to report high TF rates [14, 15]. In contrast, studies deploying clinical and/or immunologic endpoints as the sole determinants of TF tend to present the converse [14]. As VF tends to precede clinical and immunological failure by approximately 12 months [9, 19], it's a more sensitive prognosticator of TF [21]. In this cohort, the frequencies of clinical and immunological failure were, indeed, low (19 (6.8%) immunological, and 82 (29.3%) clinical failure). Therefore, the limited and sporadic use of VL testing (particularly in the period preceding 2017) adds the possibility that this study may have underestimated the frequency of TF in this cohort. This, without a doubt, highlights the importance of expediting the ongoing efforts to scale up VL testing in the country.

Aside from the relatively high frequency of TF reported in this study; the data demonstrated that TF was associated with physician documentation of suboptimal adherence; cART backbone other than AZT+3TC: (Abacavir(ABC)+(3TC); Stavudine(d4T)+3TC or Tenofovir (TDF)+Emtricitabine (FTC)); severe immunosuppression; weight-for-age and height-for-age Z-scores; cART drug substitution/holding regimens; late cART initiation calendar years and older age at cART initiation. These findings harmonize well with previous reports [21]. For

**Table 2. Crude incidence and associated factors of therapy failure among children and adolescents in National Referral Pediatric Follow-up Clinic, Asmara, Eritrea (2005–2020).**

| Characteristics[†] | Person time (years) | Events | Crude Incidence Rate (95% CI) | Rate Ratio (95% CI) | p-value[‡] |
|---|---|---|---|---|---|
| **Gender** | | | | | |
| Male | 2203.3 | 156 | 7.1 (6.1–8.3) | 1 (Ref) 0.84 (0.65–1.1) | 0.07 |
| Female | 2051.41 | 120 | 5.8 (4.9–7) | | |
| **Year of birth** | | | | | |
| ≤ 2003 | 2602.8 | 175 | 6.7 (5.8–7.8) | 1 (Ref) 0.91 (0.7–1.2) | 0.24 |
| 2004+ | 1628.8 | 100 | 6.1 (5–7.5) | | |
| **Address** | | | | | |
| Maekel | 3270.3 | 197 | 6 (5.2–6.9) | 1 (Ref) 1.3 (1–1.8) | **0.01** |
| Outside Maekel | 984.4 | 79 | 8 (6.4–10) | | |
| **Cohort Year** | | | | | |
| ≤ 2010 | 2685 | 144 | 5.4 (4.5–6.3) | 1 (Ref) 1.6 (1.2–2) | **< 0.001** |
| 2011+ | 1546.6 | 131 | 8.4 (7.1–10) | | |
| Clinical stage[§] | | | | | |
| Early | 1615.6 | 92 | 5.6 (4.6–7) | 1 (Ref) 1.2 (1–1.6) | **0.05** |
| Late | 2592.4 | 180 | 6.9 (6–8) | | |
| **Immunosuppression** | | | | | |
| Mild/Advanced | 1708.5 | 99 | 5.8 (4.7–7) | 1 (Ref) 1.2 (0.9–1.5) | 0.1 |
| Severe | 2007.3 | 136 | 6.8 (5.7–8) | | |
| **Weight for age** | | | | | |
| z > -2 | 1405.9 | 70 | 5 (3.9–6.3) | 1 (Ref) 1.5 (1.1–1.9) | **0.003** |
| z ≤ -2 | 2848.8 | 206 | 7.2 (6.3–8.3) | | |
| **Height for age** | | | | | |
| z > -2 | 1514.8 | 82 | 5.4 (4.4–6.7) | 1 (Ref) 1.3 (1–1.7) | **0.02** |
| z ≤ -2 | 2739.8 | 194 | 7.1 (6.2–8.2) | | |
| **Weight for height** | | | | | |
| z > -2 | 3473.2 | 218 | 6.3 (5.5–7.2) | 1 (Ref) 1.2 (0.9–1.6) | 0.13 |
| z ≤ -2 | 781.5 | 58 | 7.4 (5.7–9.6) | | |
| **cART Backbone** | | | | | |
| AZT + 3TC | 3615 | 216 | 6 (5.2–6.8) | 1 (Ref) 1.5 (1.1–2.1) | **0.002** |
| AZT+3TC, ELSE[¶] | 639.7 | 60 | 9.4 (7.3–12.1) | | |
| **NNRTI** | | | | | |
| EFV | 1819 | 128 | 7 (5.9–8.4) | 1 (Ref) 0.9 (0.7–1.1) | 0.1 |
| NVP | 2424.3 | 145 | 6 (5.1–7) | | |
| **Sub-optimal adherence record** | | | | | |
| No | 3705 | 75 | 5.4 (4.7–6.2) | 1 (Ref) 2.5 (1.9–3.2) | **< 0.001** |
| Yes | 549.7 | 201 | 13.6 (10.9–17.1) | | |

Abbreviations: AZT, zidovudine; 3TC, lamivudine; cART, combined antiretroviral therapy; CI, confidence interval; EFV, efavirenz; LPV/r, lopinavir/ritonavir; NNRTI, Non-nucleoside Reverse Transcriptase Inhibitors; NVP, nevirapine; TB, tuberculosis; Z-score, NCHS standard deviation.

Superscripts

[†]**Characteristics were evaluated at baseline, unless otherwise specified**

[‡]**Compares the difference of crude incidence of failure per patients' characteristics'**

[§]**WHO clinical early and advanced refer to stage 1 & 2 and stages 3 &4, respectively**

[¶]**abacavir + lamivudine, stavudine + lamivudine or tenofovir + emtricitabine.**

example, multiple studies in the region have shown that chronic malnutrition, low CD4[+] cell count, suboptimal adherence, and cART drug substitution are important drivers of first-line

**Table 3. Characteristics of the study participants stratified by therapy outcome in National Referral Pediatric Follow-up Clinic, Asmara, Eritrea (2005–2020).**

| Characteristics[†] | Therapy failure cases n = 279 | No Therapy failure n = 445 | *p*-value[‡] | Total n (%) |
|---|---|---|---|---|
| **Gender** | | | | |
| Males | 157 (41.1) | 225 (58.9) | 0.134 | 382 (52.8) |
| Females | 122 (35.7) | 220 (64.4) | | 342 (47.2) |
| **Year of birth** | 2002 (1999–2005) | 2003 (1999–2006) | 0.114 | |
| **Address** | | | | |
| Maekel | 199 (37.5) | 332 (62.5) | 0.27 | 531 (73.3) |
| Outside Maekel | 80 (41.5) | 113 (58.5) | | 193 (26.7) |
| **Age at cART initiation (months)** | 103.9 (98.4–109.5) | 99.3 (94.8–103.7) | 0.59 | |
| **Duration enrollment to cART initiation (months)** | 6 (1–33) | 7 (1–36) | 0.77 | |
| **cART initiation year** | 2010 (2008–2014) | 2011 (2008–2014) | 0.71 | |
| 2005–2009 | 109 (38.4) | 175 (61.6) | 0.87 | 284 (39.3) |
| 2010–2014 | 110 (39.7) | 167 (60.3) | | 277 (38.4) |
| 2015–2019 | 60 (39.3) | 101 (62.7) | | 161 (22.3) |
| **Clinical stage[§]** | | | | |
| Stage 1 and 2 | 94 (32.2) | 198 (67.8) | **0.005** | 290 (40.6) |
| Stage 3 and 4 | 181 (42.7) | 243 (57.3) | | 425 (59.4) |
| **TB status** | | | | |
| Not symptomatic | 242 (38.1) | 393 (61.9) | 0.28 | 634 (97.5) |
| Symptomatic | 4 (25) | 12 (75) | | 16 (2.5) |
| **Immunosuppression** | | | | |
| Mild | 30 (34.5) | 57 (65.5) | 0.06 | 87 (14.2) |
| Advanced | 70 (33.8) | 137 (66.2) | | 207 (33.8) |
| Severe | 138 (43.1) | 181 (56.9) | | 319 (52) |
| **Weight for age, z-score** | | | | |
| Z≤ -2 | 117 (37.9) | 192 (62.1) | 0.07 | 309 (67.8) |
| Z > -2 | 43 (29.3) | 104 (70.7) | | 147 (32.2) |
| **Height for age, z-score** | | | | |
| Z ≤-2 | 193 (41.2) | 276 (58.8) | **0.043** | 469 (67.5) |
| Z > -2 | 75 (33.2) | 151 (66.8) | | 226 (32.5) |
| **Weight for height, z-score** | | | | |
| Z ≤ -2 | 17 (42.5) | 23 (57.5) | 0.16 | 40 (25.8) |
| Z > -2 | 35 (30.4) | 80 (69.6) | | 115 (74.2) |
| **cART Backbone** | | | | |
| AZT + 3TC | 217 (40) | 326 (60) | 0.14 | 543 (75.1) |
| AZT + 3TC, ELSE[¶] | 61 (33.9) | 119 (66.1) | | 180 (24.9) |
| **NNRTI/PI** | | | | |
| NVP | 146 (39.9) | 220 (60.1) | 0.126 | 366 (50.6) |
| EFV | 130 (36.6) | 225 (63.4) | | 355 (49.1) |
| LPV/r | 2 (100) | 0 | | 2 (0.3) |
| **Suboptimal adherence** | | | | |
| Yes | 76 (70.7) | 30 (28.3) | **<0.001** | 106 (14.6) |
| No | 203 (32.8) | 415 (67.2) | | 618 (85.4) |
| **cART Substitution** | | | | |
| Yes | 243 (41.4) | 344 (58.6) | **0.002** | 586 (80.9) |
| No | 32 (24.8) | 97 (75.2) | | 129 (17.8) |
| Unknown | 4 (44.4) | 5 (55.6) | | 9 (1.2) |

(*Continued*)

**Table 3.** (Continued)

| Characteristics[†] | Therapy failure cases n = 279 | No Therapy failure n = 445 | p-value[‡] | Total n (%) |
|---|---|---|---|---|
| Frequency of cART change | 2 (2–3) | 1 (1–3) | < 0.001 | |

**Abbreviations**: ABC, abacavir; AZT, zidovudine; cART, combined antiretroviral therapy, CI, confidence interval; d4T, Stavudine; EFV, Efavirenz; IQR, interquartile range; LPV/r; NVP, nevirapine; TB, tuberculosis; 3TC, lamivudine; Z-score, NCHS standard deviation. P values refer to differences between included and excluded patients on baseline characteristics

Superscripts

[†]Presented as n (%) for categorical data and median (interquartile range) for continuous data

[‡] The comparisons were performed using Pearson's Chi-square test or Fisher's exact test, as appropriate, for categorical data, and Wilcoxon rank sum/Mann Whitney U-test for continuous data

[§]WHO clinical early and advanced refer to stage 1 & 2 and stages 3 &4, respectively.

[¶]abacavir + lamivudine, stavudine + lamivudine or tenofovir + emtricitabine.

**Description:** Frequency of cART change: Change constitutes changes of cART regimen among the first line options due a reason other than therapy failure e.g., due to toxicity, drug-drug interaction, among others.

cART failure [8, 14, 15]. Our finding that a higher likelihood of first-line cART failure in children who had frequent cART drug substitution is also consistent with others studies [14, 22]. Delays in pill pick-ups, distances to treatment centers, drug stock-outs, and inefficiencies in supply chains have been associated with frequent cART drug substitutions [5] and maybe decisive in this jurisdiction. In another drug-resistance mutations research, they noted that malnutrition is characterized by perverse alterations in body composition and metabolic dysfunction and that these factors may undermine the efficacy of cART [23]. Importantly, children with severe immunosuppression and/or malnourished are more susceptible to gastrointestinal infections (chronic diarrhea), possibly impeding the absorption of cART [23].

Similarities aside, country-level analysis reveals important differences between this study and other studies from the region. For instance, when years of treatment initiation were

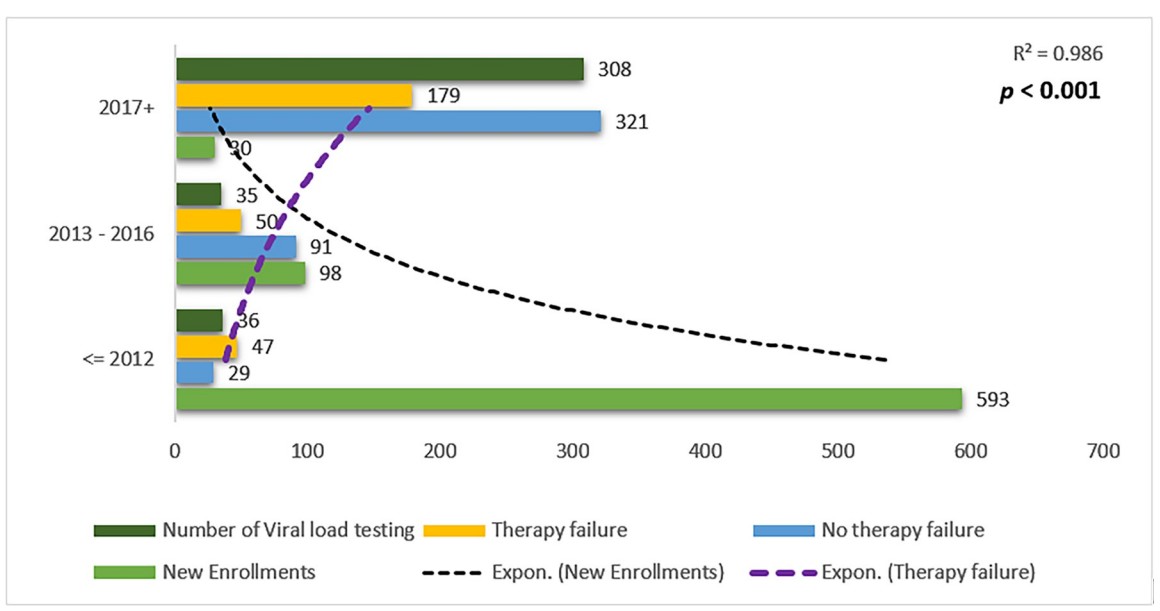

**Fig 2. Relationship among viral load tests performed in the clinic and therapy failure detection in National Referral Pediatric Follow-up Clinic, Asmara, Eritrea (2005–2020).**

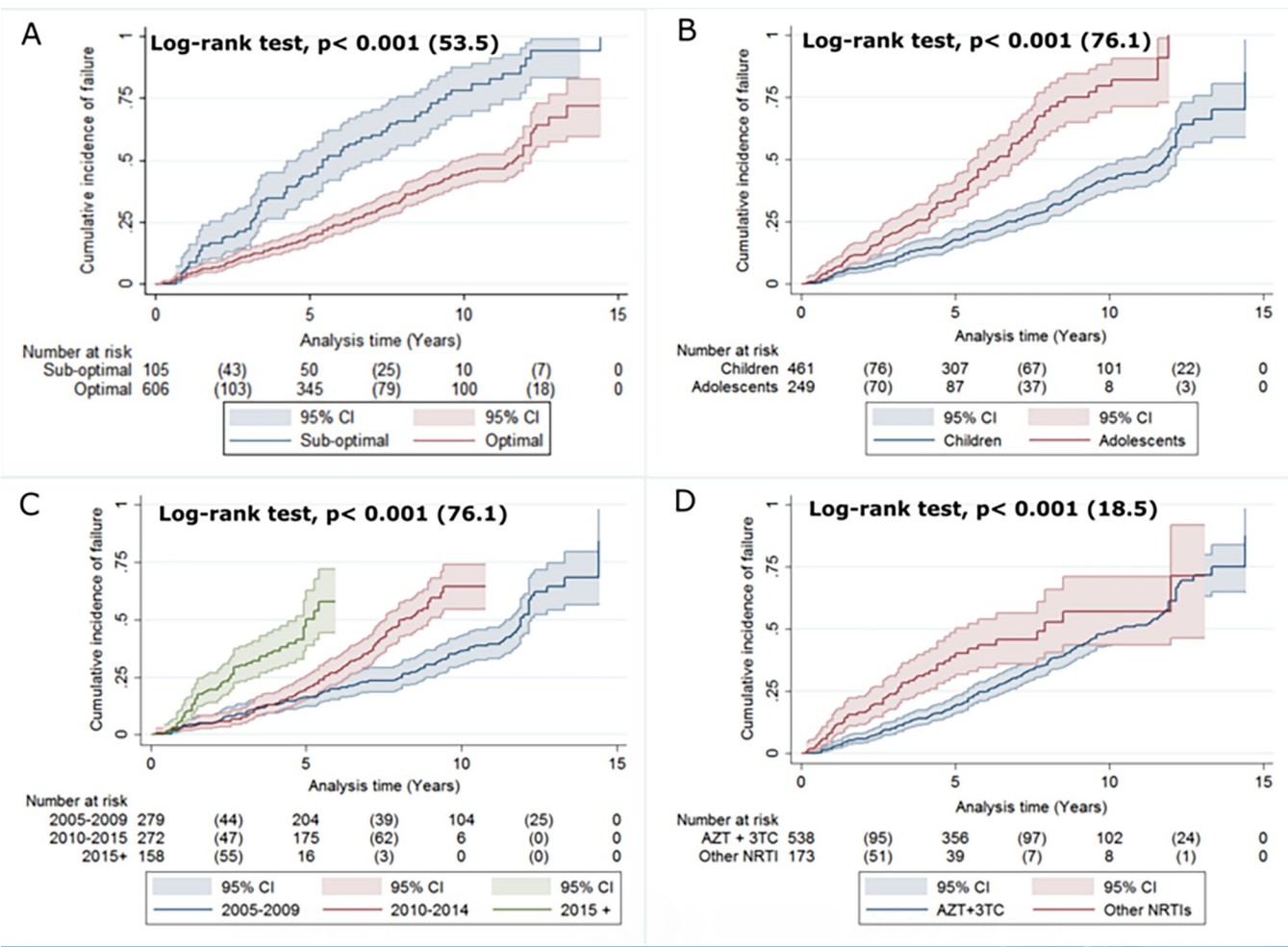

**Fig 3.** Kaplan-Meier cumulative incidence failure unadjusted curves for the pediatric cohort (n = 724) in National Referral Pediatric Follow-up Clinic by (A) Clinic's record of suboptimal adherence during the entire cohort; (B) Age at cART initiation in months; (C) cART started cohort year; and (D) Initially used c ART backbone. Log-rank p-value with chi-square is used to see the significance of differences in the Kaplan-Meier failure curves. **Definition:** Other NRTIs: abacavir + lamivudine, stavudine + lamivudine or tenofovir + emtricitabine. Adolescents are individulas whose age is greater than 10 while children are less or equal to ten.

disaggregated to 2005–2009, 2010–2014, 2015–2019 recent cART initiation was strongly associated with TF. This finding and the data demonstrating that the use of backbones other than 3TC and AZT or that older age at cART initiation is associated with TF are either unique or more common in this setting [8]. The association between the late cART initiation calendar year and TF is probably linked to the modest expansion of VL testing services in the country. Presumably, there is a linear relationship between expanded VL testing and enhanced detection of VF (Fig 2). This finding and the data showing a high failure rate of backbones other than 3TC and AZT; may point at a potential reduction in the efficacy of existing cART due to the emergence of resistance-associated mutations (RAM) as a result of prolonged usage or pre-existence of RAM before initiation of treatment-common NRTI based mutations include M184V, K65R, and four major NNRTI based mutations: 103N, Y181C, G190A, and V106M. These are well-documented possibilities [24]. To address this concern, particularly when Pre-treatment drug-resistance mutations (PDRMs) are ≥10%, the WHO recommends testing for PDRMs before initiating cART or when considering a programmatic switch of 1st-line cART

**Table 4. Cox proportional hazards of cART therapy failure among CLHIV and adolescents in National Referral Pediatric Follow-up Clinic, Asmara, Eritrea (2005–2020).**

| Characteristics[†] | Crude HR (95% CI) | *p*-value | Adjusted HR (95% CI) | *p*-value |
|---|---|---|---|---|
| **Gender** | | | | |
| Male | 1.0 (Reference) | | | |
| Female | 0.90 (0.70–1.30) | 0.8 | | |
| **Address** | | | | |
| Maekel | 1.0 (Reference) | | 1.0 (**Reference**) | |
| Outside Maekel | 1.30 (0.90–1.60) | 0.15 | 1.30 (0.90–1.70) | 0.09 |
| **Age at cART initiation** | 1.01 (1.0–1.02) | **<0.001** | 1.01 (1.0–1.02) | **<0.001** |
| **cART started calendar year** | 1.16 (1.10–1.20) | **<0.001** | 1.15 (1.10–1.30) | **<0.001** |
| **Clinical Stage** | | | | |
| Early | 1.0 (Reference) | | | |
| Advanced | 1.13 | 0.4 | | |
| **Immunosuppression** | | | | |
| Mild | 1.0 (Reference) | | 1.0 (**Reference**) | |
| Advanced | 1.01 (0.64–1.60) | 0.9 | 1.02 (0.60–1.60) | 0.9 |
| Severe | 1.40 (0.95–2.30) | 0.08 | 1.50 (1.02–2.40) | **0.04** |
| **Height for age, z score** | | | | |
| Z >-2 | 1.0 (Reference) | | | |
| Z <-2 | 1.14 (0.8–1.1.5) | 0.4 | | |
| **Weight for height, z score** | | | | |
| Z >-2 | 1.0 (Reference) | | 1.0 (**Reference**) | |
| Z <-2 | 1.50 (1.05–2.10) | **0.02** | 1.50 (1.10–2.10) | **0.02** |
| **cART Backbone** | | | | |
| AZT + 3TC | 1.0 (Reference) | | 1.0 (**Reference**) | |
| AZT+3TC, Else[¶] | 1.60 (1.10–2.40) | **0.008** | 1.60 (1.10–2.20) | **0.01** |
| **NNRTI/PI** | | | | |
| EFV | 1.0 (Reference) | | | |
| NVP | 1.16 (0.90–1.60) | 0.3 | | |
| **cART change frequency** | 0.90 (0.67–1.19) | 0.46 | | |
| **Suboptimal adherence** | | | | |
| No | 1.0 (Reference) | | 1.0 (Reference) | |
| Yes | 2.90 (2.10–3.90) | **<0.001** | 2.90 (2.20–3.90) | **<0.001** |

**Abbreviations**: AZT, zidovudine; 3TC, lamivudine; CI, confidence interval; cART, combined antiretroviral therapy; EFV, efavirenz; NVP, nevirapine; d4T, Stavudine; EFV, Efavirenz; z-scores NCHS standard deviations.

**Superscripts**

[†]**Characteristics were evaluated at baseline, unless otherwise specified**

[‡] **The analyses were performed using Cox proportional hazards model**

[§] **WHO clinical early and advanced refer to stage 1 & 2 and stages 3 &4**

[¶]**abacavir + lamivudine, stavudine + lamivudine or tenofovir+emtricitabine**

from NNRTI-based regimens. In the absence of RAM, these possibilities are hard to verify. Overall, the finding underscores the importance of research on RAM and its effect on virologic trajectories or overall treatment outcomes. In Eritrea, a previous (2016/17) unpublished survey conducted among ART initiators, estimated that PDRMs to NNRTI in adults was 7.1% (95% CI: 3.8–12.9%)–the most frequently observed mutation was in the K103 position (National CDC). However, data RAM or PDRMs in children on cART is missing. Therefore, the contribution of RAM or PDRM to TF in the country is hard to discern. Altogether, the finding

underscores the importance of research on RAM and its effect on Virologic trajectories or overall treatment outcomes.

Another result was the observed link between older age at cART initiation and TF. According to several SSA studies, younger children have a higher likelihood of VF compared to their older counterparts [8, 14]. In contrast, our data demonstrate that children who started treatment when they were older had a higher likelihood of TF. Multiple explanations can be invoked to explain this outcome including the possibility that HIV is still a stigma-laden disease in Eritrea. Difficulties facing adolescents in disclosing HIV status or caregivers/families in disclosing HIV status to their child may undermine enrolment in cART programs. Often, contact with clinicians is prompted by the recurrence of opportunistic infections or severe clinical problems. A consequence of this development is a bias towards more advanced diseases, high VL, or WHO stage-four disease [11, 25]. The data also hints at the residual effect of low attendance of pre-natal clinics by pregnant mothers in previous years. Along with expanding PMTCT coverage and pediatric DNA-based HIV-1 testing, testing strategies for children outside of these programs should be developed.

As discussed above, drug failure in this cohort is associated with several modifiable risk factors. These include malnutrition, suboptimal adherence, and delays in drug switching, or high frequency of drug substitution. According to some studies, optimization of adherence, following the expansion of virologic testing, should be the first approach to addressing TF, when resistance assays are unavailable [9]. In turn, adherence is associated with a variety of factors including drug, social-cultural background, health workers' and health system factors [8]. Another problem is the fact that physicians have only limited tools to perform reliable diagnoses of poor adherence to cART. Intervention must therefore be multi-pronged and data-driven. For instance, some possible interventions include expansion of access to alternative regimens, fixed-dose combination tablets, and syrups, strengthening drug delivery chains, patient counseling, reduction in contact intervals between patients and clinicians (weekly or bi-weekly) and following missed clinic visits by home visits.

Separately, we evaluated the duration between diagnosis of TF and switching. According to our result, a large number (68.4%) of children with TF were still on a first-line regimen. A relatively long median time to cART switch (19 months (IQR 11.3–49.7)) was also uncovered. Remarkably, long lag-time between diagnosis of TF and switch to second-line regimens is a common phenomenon in many cART programs in SSA [2, 9, 22]. According to a recent WHO report, the continued use of Nevirapine-based regimens despite the high levels of PDRMs to NNRTIs contribute to lower viral suppression among children [24]. There are multiple reasons why clinicians may delay switching. These include unavailability of generic second-line or third-line cART; the complexity of second-line options (particularly appropriate pediatric formulations); concerns regarding adherence and the need to salvage therapy in the face of emerging RAM, among others [2]. The limited range of second-line treatment options, the predominant use of clinical failure as a switch trigger, and adverse drug reactions is a plausible explanation of the reluctance by clinicians in this cohort to switch regimens.

Admittedly, the consequences of modest switching delays remain controversial. According to some investigators, remaining on failing NNRTI-based cART is associated with a heightened risk of drug resistance particularly, thymidine-associated mutations (TAMs) [2, 26]. In contrast, the ARROW study suggested that delays in the switching of up to 2 years may have limited clinical implications [27]. Similar results were reported in the CHER trial where 84% had VL<400 c/mL at the end of 5 years, with only 2.05% completing follow-up switching [28]. The possibility that endpoints of remaining on failing regimens may differ depending on the regimen, has also been suggested [22]. Although these findings have interesting implications for treatment in resource-limited settings such as Eritrea; guidelines suggest that children with

VF should be switched promptly to treatment regimens that include pharmacologically boosted PIs, or other drug combinations preferably integrase strand transfer inhibitors (INSTIs) based combinations [5, 22, 29].

## 5. Conclusion

The catalog of factors uncovered in this report bears a close resemblance to previous reports from the region. In this regard, they highlight existing concerns about the effectiveness of the current pediatric cART program in SSA. Moving forward, widespread virologic testing, prompt switching of regimens, limiting treatment interruptions, better support systems for adherence, and resistance surveillance should be emphasized. Moreover; the integration of nutritional support to tackle the impact and high magnitude of undernutrition in this population should be worked upon. Altogether, it's our considered opinion that implementing these measures is crucial in the country's pursuit of the UNAIDS 90-90-90 goals.

This study has several strengths as well as limitations. The relatively long duration of follow-up (15 years), large study population, and robust clinical data are major strengths. Nevertheless, it has several limitations. First, we may have underestimated the incidence of TF because not all patients performed VL tests. Second, retrospective studies are associated with missing covariate data. Further, critical socioeconomic data on parents or guardians and programmatic data were not collected. The approach used for adherence may also be limiting.

## Supporting information

**S1 File.**
(SAV)

## Acknowledgments

The authors would like to thank the clinical staff who supported this work at Orotta Pediatric National Referral Hospital. We are grateful to National Communicable Disease Control Division, Eritrean Ministry of Health, and ART Health Management Information System (HMIS) developers and technicians. We also thank the families and participants in the study. Our sincere appreciation goes to Dr. Ariam Mebrahtu, Dr. Natnael Belay, Dr. Yafet Tekle, Dr. Yonathan Tesfaldet, and Dr. Simon Tesfay who helped us in data collection.

## Author Contributions

**Conceptualization:** Samuel Tekle Mengistu.

**Data curation:** Samuel Tekle Mengistu, Ghirmay Ghebrekidan Ghebremeskel, Miriam Berhane Abrehe, Samuel Fisseha Tewelde.

**Formal analysis:** Samuel Tekle Mengistu, Ghirmay Ghebrekidan Ghebremeskel, Oliver Okoth Achila.

**Investigation:** Samuel Tekle Mengistu, Mahmud Mohammed Idris, Tsegereda Gebrehiwot Tikue.

**Methodology:** Samuel Tekle Mengistu, Araia Berhane Mesfin.

**Project administration:** Samuel Tekle Mengistu, Ghirmay Ghebrekidan Ghebremeskel, Oliver Okoth Achila, Araia Berhane Mesfin.

**Resources:** Samuel Tekle Mengistu, Ghirmay Ghebrekidan Ghebremeskel, Oliver Okoth Achila, Araia Berhane Mesfin.

**Software:** Samuel Tekle Mengistu.

**Supervision:** Samuel Tekle Mengistu, Ghirmay Ghebrekidan Ghebremeskel, Oliver Okoth Achila.

**Writing – original draft:** Samuel Tekle Mengistu, Ghirmay Ghebrekidan Ghebremeskel, Oliver Okoth Achila.

**Writing – review & editing:** Samuel Tekle Mengistu, Ghirmay Ghebrekidan Ghebremeskel, Oliver Okoth Achila, Miriam Berhane Abrehe, Samuel Fisseha Tewelde, Mahmud Mohammed Idris, Tsegereda Gebrehiwot Tikue, Araia Berhane Mesfin.

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
