## [Decision Letter · Decision Letter 0]

27 Apr 2022

PONE-D-21-19949

Prevalence and predictors of pediatric HIV therapy failure in a tertiary hospital in Asmara, Eritrea: A 15-year retrospective cohort study

PLOS ONE

Dear Dr. Mengstu,

Thank you for submitting your manuscript to PLOS ONE. After careful consideration, we feel that it has merit but does not fully meet PLOS ONE’s publication criteria as it currently stands. Therefore, we invite you to submit a revised version of the manuscript that addresses the points raised during the review process.

The manuscript has been evaluated by one reviewer, and his comments are available below.

The reviewer has raised a number of major concerns. He requests improvements to the reporting of methodological aspects of the study, for example, regarding the exclusion criteria of participants and the variables definition.  The reviewer also requests revision to the introduction and discussion.

Could you please carefully revise the manuscript to address all comments raised?

Please note that we have only been able to secure a single reviewer to assess your manuscript. We are issuing a decision on your manuscript at this point to prevent further delays in the evaluation of your manuscript. Please be aware that the editor who handles your revised manuscript might find it necessary to invite additional reviewers to assess this work once the revised manuscript is submitted. However, we will aim to proceed on the basis of this single review if possible. 

We look forward to receiving your revised manuscript.

Kind regards,

Lorena Verduci

Staff Editor

PLOS ONE

Journal Requirements:

3. Please upload a new copy of Figure 3 as the detail is not clear. Please follow the link for more information: https://blogs.plos.org/plos/2019/06/looking-good-tips-for-creating-your-plos-figures-graphics/" https://blogs.plos.org/plos/2019/06/looking-good-tips-for-creating-your-plos-figures-graphics/

Reviewers' comments:

Reviewer's Responses to Questions

**Comments to the Author**

1. Is the manuscript technically sound, and do the data support the conclusions?

Reviewer #1: Partly

2. Has the statistical analysis been performed appropriately and rigorously? 

Reviewer #1: No

3. Have the authors made all data underlying the findings in their manuscript fully available?

Reviewer #1: Yes

4. Is the manuscript presented in an intelligible fashion and written in standard English?

Reviewer #1: Yes

5. Review Comments to the Author

Reviewer #1: Summary of the research

Pediatric HIV remains a concern, particularly in sub-Saharan setting. The authors addressed the incidence and correlates of antiretroviral treatment failure defined by virologic, immunologic or clinical criteria. The crude incidence of failure was 6.5 events per 34 100- person-years (95% CI 5.8-7.3). Factors associated with TF were treatment adherence assess by pills count, ART backbone other than AZT and 3TC, severe immune, suppression, wasting, late ART initiation in calendar years and older age at ART initiation. Study data are important for the medical care of infants with HIV. I have some comments on the paper. These are provided below, split into minor and major.

Minor comments

The introduction and discussion part may be shortened.

Line 52: Space between 2018 and reference 1

Line 59: Do authors mean prevalent cases instead of prevalence.

Line 95: Provide ethical approval number.

Line 100: The study includes children and adolescents. Please specify this .

Line 103: Is LPV/r not a recommended first line ART treatment in children in Eritrea under 3 years?

Line 130: Please define SD

Line 167: Please explain why you provide a p-value without any comparison.

Table 4: For some odds ratios, you presented decimal number with one figure after the decimal others two, plesese present them the same way.

Line 384: The limitation section should be in the discussion section.

Major comments

- You excluded 12% of your cohort without any justification. Can you provide the details why they were excuded. As stated in table 1, excluded participants were mostly in advanced stage of the disease and lived outside of Maekel. The exclusion of these participants is source of selection bias which is not addressed in the manuscript. For the survival analyze , we suggest you include all participants and censored those lost to follow up. In addition, presenting percentages in line makes it difficult to read table 1. It would be better to present the percentages in colons.

- The study design is a retrospective study yet you report TF prevalence. In retrospective studies you can not evaluate the prevalence . You also reported a cumulative incidence and incidence rate. Can the authors clarify this in the manuscript.

- Adherence was assessed upon each follow-up visit based on missed doses. This variable changes with time and so this should be mentionned in the model. In addition, this variable was operationally defined as a categorial variable, but in table 4 it was instead used as a quantitative variable.

- The denominator of TF is not clear from line 184 to line 187 ( how did you calculate the percentages)?. In table 2 there is no reference for the rate ratio.

- The bivariate analysis in table 3 is not appropriate. Please remove this section.

- ART change during follow up is probably related to TF or vice versa, so it would not be appropriate to use this variable as a predictor.

6. PLOS authors have the option to publish the peer review history of their article (what does this mean?). If published, this will include your full peer review and any attached files.

Reviewer #1: No

---

## [Author Response · Author response to Decision Letter 0]

23 Jun 2022

We would like to thank Academic Editor of PLOS ONE and our reviewer for their invaluable inputs and constructive comments that are helpful to massively improve the quality of the manuscript. After careful consideration of the points raised, the point-by-point response are as follows:

Minor Points raised or amendments requested by reviewer #1 Authors’ Response are beneath of each reviewer's comments

2. Line 52: Space between 2018 and reference 1 

The comment has been addressed 

3. Line 59: Do authors mean prevalent cases instead of prevalence. 

The statement has been overhauled to “In Eritrea, a 2019 Spectrum modeling estimated that the magnitude of people living with HIV/AIDS (PLWHA) is 14, 000 (0.36%). Among these, children (<15 years) make up 4% and 8,956 (73%) patients are currently cART.”

4.Line 95: Provide ethical approval number 

Directive noted and the number is provided.

5.Line 100: The study includes children and adolescents. Please specify this. 

The comment has been addressed and further description of the age breakdown is stated in the methodology section called operational definition. 

6. Line 103: Is LPV/r not a recommended first line ART treatment in children in Eritrea under 3 years? 

Protease inhibitors (LPV/r) are second line regimens in Eritrea as per local guidelines due the absence of syrup based formulations. So the LPV/r tablets are similarly used as second-line regimens as in adults.

7.Line 130: Please define SD. 

SD is defined in the beginning of the statement as “Standard deviation (SD)”

8. Line 167: Please explain why you provide a p-value without any comparison. 

The comparison is now being placed and mainly compares the included vs excluded cases.

9. Table 4: For some odds ratios, you presented decimal number with one figure after the decimal others two, please

Present them the same way. 

The comment has been addressed.

10. Line 384: The limitation section should be in the discussion section 

Necessary changes have been undertaken. 

11.Discussion may be shortened 

Some sentences have been deleted. 

Major Revision

Major points raised or amendments requested by Reviewer #1 Authors’ response

1. You excluded 12% of your cohort without any justification. Can you provide the details why they were excluded? As stated in table 1, excluded participants were mostly in advanced stage of the disease and lived outside of Maekel. The exclusion of these participants is source of selection bias which is not addressed in the manuscript. 

The detailed justification for exclusion are specified in figure 1. 97 cases were excluded out of which, 54 patients were excluded due to follow-up less than 6 months. 

In studies of this kind [See Costenaro P, Penazzato M, Lundin R, Rossi G, Massavon W, Patel D, et al. Predictors of Treatment Failure in HIV-Positive Children Receiving Combination Antiretroviral Therapy : Cohort Data From Mozambique and Uganda. 2015;4(1):39–48.], treatment response is evaluated at 6 months following treatment initiation. The reasoning behind this cut-off is that treatment response cannot be defined prior to this duration. An additional 42 cases were excluded due to key missing follow up data: unknown follow up duration and those with no therapy outcome endpoints. Missing data is a common problem in retrospective studies of this kind. Consequently, this statement has been in cooperated in the manuscript. 

2. For the survival analysis, we suggest you include all participants and censored those lost to follow up. 

This has been addressed in the methodology subsection, end-point definition as “For TF analysis, the period of follow-up was from cART initiation up to the earliest detection of TF. Children without TF were censored at the date of death, lost to follow-up (defined as missing follow-up visits for more than 6 months), transferred to another clinic, or the date record of any last event in the clinic.” So, yes we included all participants and censored those lost to follow up.

3. Presenting percentages in line makes it difficult to read table 1. It would be better to present the percentages in colons. 

The comment has been addressed. All the percentages are across columns.

4. The study design is a retrospective study yet you report TF prevalence. In retrospective studies you cannot evaluate the

Prevalence. You also reported a cumulative incidence and incidence rate. Can the authors clarify this in the manuscript? 

From epidemiologic literatures (Biostatistics and epidemiology: a primer for health and biomedical prefessionals / by SylviaWassertheil-Smoller. —3rd ed.page 92-93), our understanding of prevalence is:

= Number of persons with a condition / Total number of persons in population at risk for the condition at a particular point in time

In our case, our population is children living with HIV receiving cART in the study setting and the condition is therapy failure. 

On the other hand, incidence means:

= Number of new cases of a disease per unit of time / Total number at risk in beginning of this time period 

In our opinion, so long as patient’s information is well documented, period prevalence of a specific disease in a population can be calculated. 

Overall, it our submission that period prevalence can be calculated using retrospective data so long as the data is well collected and the time period is specified.

The below listed references used similar approach to determine the burden (prevalence) of TF.

A. Solomon Weldegebreal Asgedom et al., Immunologic and Clinical Failure of Antiretroviral Therapy in People Living with Human Immunodeficiency Virus within Two Years of Treatment. Hindawi BioMed Research International Volume 2020, Article ID 5474103, 8 pages

B. Dow et al.: Durability of antiretroviral therapy and predictors of virologic failure among perinatally HIV-infected children in Tanzania: a four-year follow-up. BMC Infectious Diseases 2014 14:567

C. Isaac O. Abah et al: Antiretroviral Therapy-associated Adverse Drug Reactions and their Effects on Virologic Failure- A Retrospective Cohort Study in Nigeria. Current HIV Research, 2018, 16, 436-446

D. Bacha et al.: Predictors of treatment failure and time to detection and switching in HIV-infected Ethiopian children receiving first line anti-retroviral therapy. BMC Infectious Diseases 2012 12:197.

5. Adherence was assessed upon each follow-up visit based on missed doses. This variable changes with time and so this

Should be mentioned in the model. 

The definition of suboptimal adherence comprises of at least one record of fair or poor adherence. Hence patients were classified as either having a record of suboptimal adherence or no record throughout their follow up period.

6. In addition, this variable was operationally defined as a categorical variable, but in Table 4 it was instead used as a quantitative variable. 

This comment has been addressed.

7. The denominator of TF is not clear from line 184 to line 187 (how did you calculate the percentages)? 

Absolute numbers have been included. e.g. 279/724 for a prevalence of 38%. The denominator is total number ofCLHIV whereas the numerators utilized are number Virologic failure alone to calculate VF, Immunologic failure alone to calculate IF, clinical failure alone to calculate CF and number of patients who had virologic, immunologic and clinical failure at the same time.

8. In table 2 there is no reference for the rate ratio. 

This comment has been addressed.

9. The bivariate analysis in table 3 is not appropriate. Please remove this section. 

Table analyzed factors associated with the prevalence of TF. While table 2 analyses factors as per subcategories of the variables’ incidence. 

However, and based on your opinion, we can either retain the table (after revising the title – e.g we have amended the title to factors associated with prevalence TF) as is generally the case in studies of this kind or attach it as a supplementary file. 

We can also omit it altogether. 

10. ART change during follow up is probably related to TF or vice versa, so it would not be appropriate to use this variable as a predictor. We have substituted the word cART change with the word, cART substitution. cART substitution is not necessarily associated TF but can be linked to stock outs, adverse drug reactions and emerging of preferable regimens. cART switching is definitely due TF. Our variable merely captured cART substitution.

---

## [Decision Letter · Decision Letter 1]

19 Sep 2022

PONE-D-21-19949R1Prevalence and predictors of pediatric HIV therapy failure in a tertiary hospital in Asmara, Eritrea: A 15-year retrospective cohort studyPLOS ONE

Dear Dr. Mengstu,

Thank you for submitting your manuscript to PLOS ONE. After careful consideration, we feel that it has merit but does not fully meet PLOS ONE’s publication criteria as it currently stands. Therefore, we invite you to submit a revised version of the manuscript that addresses the points raised during the review process. Your revised manuscript has been assessed by three peer-reviewers, including a statistical reviewer, and their reports are appended below.  The reviewers comment that the analyses reported in the study could be improved upon and that they could be put into context rather than report risk factors only. In addition, the reviewers comment that some of the references reported in this study should be updated to reflect the, now-published, 2021 data.  Could you please revise the manuscript to carefully address the concerns raised?

We look forward to receiving your revised manuscript.

Kind regards,

Maria Elisabeth Johanna Zalm, Ph.D

Editorial Office

PLOS ONE

Reviewers' comments:

Reviewer's Responses to Questions

**Comments to the Author**

1. If the authors have adequately addressed your comments raised in a previous round of review and you feel that this manuscript is now acceptable for publication, you may indicate that here to bypass the “Comments to the Author” section, enter your conflict of interest statement in the “Confidential to Editor” section, and submit your "Accept" recommendation.

Reviewer #2: All comments have been addressed

Reviewer #3: (No Response)

Reviewer #4: (No Response)

Reviewer #5: All comments have been addressed

2. Is the manuscript technically sound, and do the data support the conclusions?

Reviewer #2: Yes

Reviewer #3: Partly

Reviewer #4: Yes

Reviewer #5: Yes

3. Has the statistical analysis been performed appropriately and rigorously? 

Reviewer #2: Yes

Reviewer #3: No

Reviewer #4: Yes

Reviewer #5: Yes

4. Have the authors made all data underlying the findings in their manuscript fully available?

Reviewer #2: Yes

Reviewer #3: Yes

Reviewer #4: Yes

Reviewer #5: Yes

5. Is the manuscript presented in an intelligible fashion and written in standard English?

Reviewer #2: Yes

Reviewer #3: Yes

Reviewer #4: Yes

Reviewer #5: Yes

6. Review Comments to the Author

Reviewer #2: The authors addressed all the comments. However, I still see a few typos: eg, line 162, something is missing between CLHIV and the next sentence; title of Table 1. Please check for similar errors.

Reviewer #3: General comments:

I had some questions about the analyses and I think they could be improved. Also, if at all possible, I encourage the authors to try to put their analyses in context better rather than just reporting risk factors. For instance, attributable risks might be nice though maybe not feasible.

Specific comments:

1. (lines 28, 251, etc.) The term "predictors" is used in this manuscript. I believe the authors are treating this as a synonym to "factors" and "independent variables" but I would encourage the authors to not use the term "predictors". The authors have not built a predictive model, they are looking at associations between these variables and the outcomes.

2. (lines 123-4) Does "persistent" mean CD4+ levels below the threshold for six months for every measurement?

3. (lines 151-156) Some of the covariates in the Cox model appear to be time-dependent. How were those handled? Would it be better to switch to a multiple endpoint Cox model, such as an Anderson-Gill model? Also, since treatment failure probably does not happen at the time of the survey, i.e., it happens between two surveys, was the outcome treated as interval censored?

4. (lines 156-158) Stepwise variable selection procedures, which include backward selection, usually do not do a good job of finding the most appropriate model (e.g., https://doi.org/10.1002/sim.3943). Stepwise procedures and any p-value based selection have quite a bit of evidence suggesting that they are poor at selecting the appropriate variables. For a decent summary, see the link above. It's better to select based on more robust criteria, especially measures which assess the fit of the model or, better yet, a penalized estimator such as lasso or lars.

5. (lines 157-158) Usually stepwise variable selection procedures select based on the p-value of the predictor variable, which is not a measure of model fit. If you have used something other than the default, that is not mentioned here. Regardless, I strongly recommend not using stepwise selection procedures.

6. (Table 1) Significance testing in these situations is generally frowned upon because a non-significant p-value does not indicate that groups are the same. For info on the topic in relation to baseline imbalance in randomized trials see Altman, https://doi.org/10.2307/2987510 and Senn, https://doi.org/10.1002/sim.4780131703. I believe the same logic extends to your table 1. My recommendation is to remove the significance testing from table 1 and use standardized difference to assess differences (see Austin, https://doi.org/10.1080/03610910902859574).

7. (Table 2) How were the CIs calculated for these rates? I don't believe I saw that in the statistical methods section.

8. (line 251) Although I know the phrase "independent predictors" is used quite a bit, I don't understand the usage of "independent" in a multivariable model since the effect is dependent on the other predictors in the model. I recommend not using that phrasing and something along the lines of "after controlling for other factors".

Reviewer #4: Thank you for the opportunity to review this paper looking at treatment failure on first line ART in Eritrea. Overall, the methods are appropriate to address the research question and results are generally clearly presented. I have several comments below – most of these individually are very minor. Main overall comment relates to the ordering of the results:

The authors look at the outcome in 3 ways – a prevalence (which does not take into account the follow up time), crude incidence rates (giving estimates of TF per PYs of follow up ) and then using KM an cox models to look at cumulative incidence of TF. I would suggest considering reordering these and presenting table 3 first. While this table provides a summary of characteristics in those who failed and those who did not, it doesn’t take into account the duration of follow up and to me this would make sense to come first. Next the crude incidence rates (table 2) could follow and then the KM and cox modelling.

Other comments:

Abstract results

Line 33 - Prevalence for TF is provided, this is the rate across all follow-up – it would be useful to also quote here the duration of follow-up. E.g., “prevalence was 38.5% () over a median follow up of XX [XX-XX]”

line 42 – missing word (seven in one hundred)

Introduction

Line 48 - UNAIDS estimates are provided from 2019 in the first paragraph. Data for 2021 have now been released – some of these references could be updated.

Line 59 – number of PLWHA in 2019 WAS 14000 (not IS, as now in the past). This figure is also quoted with a % - presumably, this is the % of the total population? This should be stated to be clear.

Line 60 – this sentence starts with providing information on children then quotes the number on ART, which appears to relate to the total population, including adults. This could be reordered as this is not immediately clear.

Line 76 – “ranging” – should be “range”

Materials and methods

Line 91 – states 822 children <15 were seen be the clinic (this is the total shown in the flow diagram). Line 100 states all individuals <18 were included. Is the criteria to be seen in the clinic being <15 years, but pts were included in the analysis as long as they initiated ART by 18 years? Or were they only followed up to 18 years? Please clarify why the age cut offs differ

Line 102 – How is it possible to have an unknown duration of follow up, and why is this treated differently from being LTFU? If a pt was seen in clinic and followed for a certain time, couldn’t they be censored at the last point they were known to be in follow up?

Line 108 – were missed ART doses self reported by the pts? This should be stated

Line 123 – ‘persistent’ should be defined. Eg, does this mean 2 CD4s below the specified levels over a certain time? When was the failure deemed to have occurred? At the time of the first measurement?

Line 128 – “<10%” should be “5 to <10%”. I note the previous reviewer also queried the inclusion of adherence in the model. If this was recorded at every follow up visit, the authors could consider inclusion as a time update variable in the Cox model, rather than the binary suboptimal adherence variable used.

Line 134 – height-for-age is missing a “-”

Line 153 – year of CART start was included in the model and not listed here

Line 155 – cART change frequency – please specify what constitutes a change.

Results

Line 169 – Before comparing those included/included it would be useful to state in text that 97 were excluded due to …..

Line 172 – authors state the majority of those included were wasted etc compared to those excluded. I think this should be “fewer” of the participants included were stunted, wasted etc. Eg only 25.6% of those included had weight for age <-2 compared to 56.5% of those excluded.

Line 188 – the median time to TF is quoted, presumably this is median time in those who failed? This should be specified to distinguish between other estimates of median time to TF estimated using KM later in the results.

Line 194 – table 2. Why do the number of events in table 2 differ from table 3? EG 156 males with TF in table 2 and 157 in table 3.

Line 204 – table 3 presents row percentages, for example showing what proportion of pt with WHO mild, moderate, and advance stage had TF. The text starting line 204 describes characteristics of those who failed compared to those who didn’t. While in the end the differences are equivalent/follow on from each other, from the text, I would expect to see column percentages in table 3 so the trends described can easily be seen (or else the text should be amended to better reflect the table format).

Line 238 – at first use should specify the ‘median’ quoted is ‘median time to failure’.

Line 240 – ‘mean’ is given for some years. Further in line 242 its referred to as ‘average’ time to failure. Consistent language should be used (and assume it should be median throughout?)

Discussion

Line 273 – the prevalence’s quoted in other countries do not read well, please review sentence structure

Line 340 – ‘delays in drug switching’ – where/how was this assessed?

Reviewer #5: In this study the authors aimed to investigate the prevalence, incidence, and predictors of first-line cART failure using the virologic (plasma viral load), immunologic and clinical criteria among children and adolescents living with HIV (CLHIV) on cART. They found that the prevalence of therapy failure was 279/724 (38.5% (95% CI 35-195 42.2) and they analyzed the factors that might have been associated with the TF. The idea to perform the study is interesting, it's methodology is correct. The number of patients is high. The manuscript is well prepared, however several minor stylistic or language corrections are required. The authors have answered all the queries raised at the first revision. Thus, I have no further queries.

7. PLOS authors have the option to publish the peer review history of their article (what does this mean?). If published, this will include your full peer review and any attached files.

Reviewer #2: No

Reviewer #3: No

Reviewer #4: No

Reviewer #5: No

---

## [Author Response · Author response to Decision Letter 1]

10 Nov 2022

See attached document - point by point response.doc

---

## [Decision Letter · Decision Letter 2]

6 Dec 2022

PONE-D-21-19949R2

Prevalence and factors associated with pediatric HIV therapy failure in a tertiary hospital in Asmara, Eritrea: A 15-year retrospective cohort study

PLOS ONE

Dear Dr. Mengstu,

Thank you for submitting your manuscript to PLOS ONE. After careful consideration, we feel that it has merit but does not fully meet PLOS ONE’s publication criteria as it currently stands. Therefore, we invite you to submit a revised version of the manuscript that addresses the points raised during the review process.

We look forward to receiving your revised manuscript.

Kind regards,

Dr. CM González-Domenech

Academic Editor

PLOS ONE

Journal Requirements:

Additional Editor Comments:

Dear Dr. Mengstu,

Thank you for submitting your reviewed manuscript to PLOS ONE. This has improved considerably addressing the suggestions and concerns from the reviewers. Anyway, one out of the three reviewers still has some comments which you can find below. In addition, I also have few just minor comments (comment 7 from Reviewer 3 related to CI calculation is not included in the Method section yet; comments 2 and 19 from Reviewer 4 were answered but not included in the corresponding sections, Method and Discussion, respectively; line 100 of reviewed version: an “and” is missing (between duration, “and” unknown); line 226: One of the p for the p-values is differently in capital letter).

If applicable, we recommend that you deposit or indicate how the raw data could be accessed, as the reviewer 4 pointed.

#Reviewer 4

The authors have addressed/responded to previous comments adequately. Below are some additional minor comments.

Authors state all relevant data are contained within the paper and supporting files. The raw data are not, so could be clarified if/how data can be accessed

Abstract

Line 22 - Typo in first line – (should be “treatment failure IN”)

Line 29 – Data are plural, so should be “Data were”

Line 41 – score is missing “weight for height z-score” . AHR has a capital A, previously was aHR

Line 45 – The authors state seven in one hundred children likely to develop TF. The results is per 100 person years – so the conclusion should clarify its each year.

Introduction

Line 54 – what year does the 53% refer to? Reference is recent UNAIDS report but previous sentence refers to 2018 so this isn't clear. And is the 53% in the focus countries, or globally?

Methods

Line 99 – the authors previously elaborated on what was meant by unknown follow up duration. However, in the manuscript I think this might still be unclear to the reader (“Additional exclusion considerations included unknown follow up duration, unknown therapy outcome endpoints (Fig 1)”. The figure describes this as “Key missing follow up data”. Should this be “Missing key follow up data”? Could you simply state in the manuscript – Unknown treatment outcome? Or even “missing key follow up data”? The issue is you don’t have the data to determine if/when the outcome occurred, but this isn’t really clear.

Line 120 – “6 months of effective treatment” – what is meant by effective? Or do you just mean after 6 months of being on treatment (as for the immunological definition).

Line 124 – does the 3 months refer to time on treatment, or the space between VLs (could be clarified in text).

Results

Line 184 – p<0.03 – should be p = 0.03

Table 1: should p for weight for age be p<0.001 instead of p<0.005?

Table 2 and 3 – the table numbers are the wrong way round in the table titles.

Line 213 – table 2 is referred to after the median time to cART switch – but that is not the information presented in the table. Table 2 should be properly introduced in the text. Eg “Crude incidence rates by key characteristics are provided in table 2”

Table 4: The authors explained in their response what was meant by cART change frequency, but I think would be helpful to include this in the manuscript – e.g. in the table footnote.

Discussion

Paragraph 1 – some studies give a time frame for duration of follow up – is it possible to say anything about the others? Its difficult to interpret rates without information on timepoint/duration

Reviewers' comments:

Reviewer's Responses to Questions

**Comments to the Author**

1. If the authors have adequately addressed your comments raised in a previous round of review and you feel that this manuscript is now acceptable for publication, you may indicate that here to bypass the “Comments to the Author” section, enter your conflict of interest statement in the “Confidential to Editor” section, and submit your "Accept" recommendation.

Reviewer #2: All comments have been addressed

Reviewer #3: All comments have been addressed

Reviewer #4: (No Response)

2. Is the manuscript technically sound, and do the data support the conclusions?

Reviewer #2: Yes

Reviewer #3: (No Response)

Reviewer #4: Yes

3. Has the statistical analysis been performed appropriately and rigorously? 

Reviewer #2: Yes

Reviewer #3: (No Response)

Reviewer #4: Yes

4. Have the authors made all data underlying the findings in their manuscript fully available?

Reviewer #2: Yes

Reviewer #3: (No Response)

Reviewer #4: No

5. Is the manuscript presented in an intelligible fashion and written in standard English?

Reviewer #2: Yes

Reviewer #3: (No Response)

Reviewer #4: Yes

6. Review Comments to the Author

Reviewer #2: All comments are adequately addressed and I am satisfied with the responses. No further comment from my end.

Reviewer #3: Thank you for your thoughtful consideration of my comments. To follow up on one of them,

1. (comment 6) Yes, I recommend removing the p-value column from table 1.

2. (comment 7) I'm not familiar enough with Stata to know how this is calculated. If possible, I recommend including a methodological citation for how the CIs were calculated. This will depend on whether the Stata help has a citation.

Reviewer #4: The authors have addressed/responded to previous comments adequately. Below are some additional minor comments.

Authors state all relevant data are contained within the paper and supporting files. The raw data are not, so could be clarified if/how data can be accessed

Abstract

Line 22 - Typo in first line – (should be “treatment failure IN”)

Line 29 – Data are plural, so should be “Data were”

Line 41 – score is missing “weight for height z-score” . AHR has a capital A, previously was aHR

Line 45 – The authors state seven in one hundred children likely to develop TF. The results is per 100 person years – so the conclusion should clarify its each year.

Introduction

Line 54 – what year does the 53% refer to? Reference is recent UNAIDS report but previous sentence refers to 2018 so this isn't clear. And is the 53% in the focus countries, or globally?

Methods

Line 99 – the authors previously elaborated on what was meant by unknown follow up duration. However, in the manuscript I think this might still be unclear to the reader (“Additional exclusion considerations included unknown follow up duration, unknown therapy outcome endpoints (Fig 1)”. The figure describes this as “Key missing follow up data”. Should this be “Missing key follow up data”? Could you simply state in the manuscript – Unknown treatment outcome? Or even “missing key follow up data”? The issue is you don’t have the data to determine if/when the outcome occurred, but this isn’t really clear.

Line 120 – “6 months of effective treatment” – what is meant by effective? Or do you just mean after 6 months of being on treatment (as for the immunological definition).

Line 124 – does the 3 months refer to time on treatment, or the space between VLs (could be clarified in text).

Results

Line 184 – p<0.03 – should be p = 0.03

Table 1: should p for weight for age be p<0.001 instead of p<0.005?

Table 2 and 3 – the table numbers are the wrong way round in the table titles.

Line 213 – table 2 is referred to after the median time to cART switch – but that is not the information presented in the table. Table 2 should be properly introduced in the text. Eg “Crude incidence rates by key characteristics are provided in table 2”

Table 4: The authors explained in their response what was meant by cART change frequency, but I think would be helpful to include this in the manuscript – e.g. in the table footnote.

Discussion

Paragraph 1 – some studies give a time frame for duration of follow up – is it possible to say anything about the others? Its difficult to interpret rates without information on timepoint/duration

7. PLOS authors have the option to publish the peer review history of their article (what does this mean?). If published, this will include your full peer review and any attached files.

Reviewer #2: No

Reviewer #3: No

Reviewer #4: No

---

## [Author Response · Author response to Decision Letter 2]

1 Jan 2023

Minor Points raised or amendments requested by Academic Editor (Dr. CM González-Domenech)

1. 1. Comment 7 from Reviewer 3 related to CI calculation is not included in the Method section yet These CI were generated by STATA v 12 thru the tab of statistics > survival analysis > Summary statistics, tests and tables > Person-time, incidence rates and SMR after defining the data in time to event form.

Generally, these are equivalent to unadjusted HR with CI calculated in regard to their respective SE.

This statement has been incorporated in the third round of revision.

2. Comments 2 and 19 from Reviewer 4 were answered but not included in the corresponding sections, Method and Discussion, respectively Will be addressed

3. Line 100 of reviewed version: an “and” is missing (between duration, “and” unknown); line 226: One of the p for the p-values is differently in capital letter. The needed amendments have been carried out.

4. If applicable, we recommend that you deposit or indicate how the raw data could be accessed, as the reviewer 4 pointed. The raw data has been depositied along with reseach materials

#Reviewer 4 Authors’ Response

Abstract 

1. Line 22 - Typo in first line – (should be “treatment failure IN”) Addressed

2. Line 29 – Data are plural, so should be “Data were” Addressed

3. Line 41 – score is missing “weight for height z-score”. AHR has a capital A, previously was aHR Addressed

4. Line 45 – The authors state seven in one hundred children likely to develop TF. The results is per 100 person years – so the conclusion should clarify its each year. Addressed

Introduction 

1. Line 54 – what year does the 53% refer to? Reference is recent UNAIDS report, but previous sentence refers to 2018 so this isn't clear. And is the 53% in the focus countries, or globally? The statement has been overhauled. The 53% represents the number of children covered by cART in focus countries. 

2. Comments 2 and 19 from Reviewer 4 were answered but not included in the corresponding sections, Method and Discussion, respectively Will be addressed

3. Line 100 of reviewed version: an “and” is missing (between duration, “and” unknown); line 226: One of the p for the p-values is differently in capital letter. The needed amendments have been carried out.

4. If applicable, we recommend that you deposit or indicate how the raw data could be accessed, as the reviewer 4 pointed. The raw data has been depositied along with reseach materials

#Reviewer 4 Authors’ Response

Abstract 

1. Line 22 - Typo in first line – (should be “treatment failure IN”) Addressed

2. Line 29 – Data are plural, so should be “Data were” Addressed

3. Line 41 – score is missing “weight for height z-score”. AHR has a capital A, previously was aHR Addressed

4. Line 45 – The authors state seven in one hundred children likely to develop TF. The results is per 100 person years – so the conclusion should clarify its each year. Addressed

Introduction 

1. Line 54 – what year does the 53% refer to? Reference is recent UNAIDS report but previous sentence refers to 2018 so this isn't clear. And is the 53% in the focus countries, or globally? The statement has been overhauled. The 53% represents the number of children covered by cART in focus countries. 

Methods 

1. Line 99 – the authors previously elaborated on what was meant by unknown follow up duration. However, in the manuscript I think this might still be unclear to the reader (“Additional exclusion considerations included unknown follow up duration, unknown therapy outcome endpoints (Fig 1)”. The figure describes this as “Key missing follow up data”. Should this be “Missing key follow up data”? Could you simply state in the manuscript – Unknown treatment outcome? Or even “missing key follow up data”? The issue is you don’t have the data to determine if/when the outcome occurred, but this isn’t really clear. Statement has been incorporated to make it clear

2. Line 120 – “6 months of effective treatment” – what is meant by effective? Or do you just mean after 6 months of being on treatment (as for the immunological definition). The word “effective” is erased as it’s a deplicate information as outcome with adherence support was already mentioned in line 117.

3. Line 124 – does the 3 months refer to time on treatment, or the space between VLs (could be clarified in text). overhauled

Results 

1. Line 184 – p<0.03 – should be p=0.03 overhauled

2. Table 1: should p for weight for age be p<0.005? overhauled

3. Table 2 and 3 – the table numbers are the wrong way round in the table titles. overhauled

4. Line 213 – table 2 is referred to after the median time to cART switch – but that is not the information presented in the table. Table 2 should be properly introduced in the text. Eg “Crude incidence rates by key characteristics are provided in table 2” True. Changes has been made

5. Table 4: The authors explained in their response what was meant by cART change frequency, but I think would be helpful to include this in the manuscript – e.g. in the table footnote. Added

Discussion 

1. Paragraph 1 – some studies give a time frame for duration of follow up – is it possible to say anything about the others? It’s difficult to interpret rates without information on time point/duration True, as the results are from different studies of different methodology, duration of follow-up couldn’t be obtained from all of them. Some are only cross-sectional, evaluating prevalence of TF while others are cohorts, evaluating both prevalence and incidences.

Reviewer #3 Authors’ Response

1. (comment 6) Yes, I recommend removing the p-value column from table 1. Its removed.

2. (comment 7) I'm not familiar enough with Stata to know how this is calculated. If possible, I recommend including a methodological citation for how the CIs were calculated. This will depend on whether the Stata help has a citation. It is now cited.

---

## [Editor Report · Decision Letter 3]

6 Jan 2023

PONE-D-21-19949R3Prevalence and factors associated with pediatric HIV therapy failure in a tertiary hospital in Asmara, Eritrea: A 15-year retrospective cohort studyPLOS ONE

Dear Dr. Samuel Tekle Mengistu,

Thank you for submitting your manuscript to PLOS ONE. After careful consideration, we feel that it has merit but does not fully meet PLOS ONE’s publication criteria as it currently stands. Therefore, we invite you to submit a revised version of the manuscript that addresses the points raised during the review process.

ACADEMIC EDITOR:

Almost all the comments from Reviewers have already been addressed but some minor mistakes still remained. Please, amend them before accepting for publication:

Cite 23. The title of the article is wrong. “No TPredicting” must be replaced by “Predicting […]”

Table 2 and 3, p-value with "V" in lowcase.

Line 40. AHR is still in capital letter (Reviewer 4´s comment not addressed).

Table 1. Label “Excluded” has dissappeared in the column but the corresponding data are remaining

Please submit your revised manuscript within one week (12th of January 2023). If you will need more time than this to complete your revisions, please reply to this message or contact the journal office at plosone@plos.org. Please include the following items when submitting your revised manuscript:A rebuttal letter that responds to each point raised by the academic editor and reviewer(s). You should upload this letter as a separate file labeled 'Response to Reviewers'.A marked-up copy of your manuscript that highlights changes made to the original version. You should upload this as a separate file labeled 'Revised Manuscript with Track Changes'.An unmarked version of your revised paper without tracked changes. You should upload this as a separate file labeled 'Manuscript'.If applicable, we recommend that you deposit your laboratory protocols in protocols.io to enhance the reproducibility of your results. Protocols.io assigns your protocol its own identifier (DOI) so that it can be cited independently in the future. For instructions see: https://journals.plos.org/plosone/s/submission-guidelines#loc-laboratory-protocols. Additionally, PLOS ONE offers an option for publishing peer-reviewed Lab Protocol articles, which describe protocols hosted on protocols.io. Read more information on sharing protocols at https://plos.org/protocols?utm_medium=editorial-email&utm_source=authorletters&utm_campaign=protocols.

We look forward to receiving your revised manuscript.

Kind regards,

Carmen María González-Domenech, PhD

Academic Editor

PLOS ONE

Journal Requirements:

Additional Editor Comments (if provided):

Almost all the comments from Reviewers have already been addressed but some minor mistakes still remained. Please, amend them before accepting for publication:

Cite 23. The title of the article is wrong. “No TPredicting” must be replaced by “Predicting […]”

Table 2 and 3, p-value with "V" in lowcase.

Line 40. AHR is still in capital letter (Reviewer 4´s comment not addressed).

Table 1. Label “Excluded” has dissappeared in the column but the corresponding data are remaining
---

## [Author Response · Author response to Decision Letter 3]

18 Feb 2023

To PLOS ONE Editorial Office

We would like to thank staff Editor of PLOS ONE and our reviewers for their invaluable inputs and constructive comments that are helpful to massively improve the quality of the manuscript. After careful consideration of the points raised, the point-by-point response to the fourth round of revision are as follows:

1. Cite 23 is misspelled: Addressed.

2. P-values in table 2 and 3 should be in low case: The letters are changed to low case now.

3. Line 40: AHR, ‘A’ still in capital letter: The term adjusted hazards ratio has been abbreviated as aHR consistently throughout the document now.

4. Table 1. The label “Excluded” has been removed but the corresponding data still exists: The table is checked for any missing labels of the column, excluded but it still holds.

---

## [Editor Report · Decision Letter 4]

21 Feb 2023

Prevalence and factors associated with pediatric HIV therapy failure in a tertiary hospital in Asmara, Eritrea: A 15-year retrospective cohort study

PONE-D-21-19949R4

Dear Dr. Mengistu,

We’re pleased to inform you that your manuscript has been judged scientifically suitable for publication and will be formally accepted for publication once it meets all outstanding technical requirements.

Kind regards,

Carmen María González-Domenech, Ph.D.

Academic Editor

PLOS ONE

---

## [Editor Report · Acceptance letter]

27 Feb 2023

PONE-D-21-19949R4 

Prevalence and factors associated with pediatric HIV therapy failure in a tertiary hospital in Asmara, Eritrea: A 15-year retrospective cohort study 

Dear Dr. Mengistu:

I'm pleased to inform you that your manuscript has been deemed suitable for publication in PLOS ONE. Congratulations! Your manuscript is now with our production department. 

Kind regards, 

on behalf of

Dr. Carmen María González-Domenech 

Academic Editor

PLOS ONE